# The Assessment of Body Composition and Nutritional Awareness of Football Players According to Age

**DOI:** 10.3390/nu15030705

**Published:** 2023-01-30

**Authors:** Wiktoria Staśkiewicz, Elżbieta Grochowska-Niedworok, Grzegorz Zydek, Mateusz Grajek, Karolina Krupa-Kotara, Agnieszka Białek-Dratwa, Sylwia Jaruga-Sękowska, Oskar Kowalski, Marek Kardas

**Affiliations:** 1Department of Food Technology and Quality Evaluation, Department of Dietetics, Faculty of Health Sciences in Bytom, Medical University of Silesia in Katowice, Ul. Jordana 19, 41-808 Zabrze, Poland; 2Department of Health Sciences and Physical Culture, University of Applied Sciences in Nysa, Ul. Ujejskiego 12, 48-300 Nysa, Poland; 3Department of Sport Nutrition, Jerzy Kukuczka Academy of Physical Education in Katowice, Ul. Mikołowska 72A, 40-065 Katowice, Poland; 4Department of Public Health, Department of Public Health Policy, Faculty of Health Sciences in Bytom, Medical University of Silesia in Katowice, Ul. Piekarska 18, 41-902 Bytom, Poland; 5Department of Epidemiology, Faculty of Health Sciences in Bytom, Medical University of Silesia in Katowice, Ul. Piekarska 18, 41-902 Bytom, Poland; 6Department of Human Nutrition, Department of Dietetics, Faculty of Health Sciences in Bytom, Medical University of Silesia in Katowice, Ul. Jordana 19, 41-808 Zabrze, Poland; 7Department of Health Promotion, Faculty of Health Sciences in Bytom, Medical University of Silesia in Katowice, Ul. Piekarska 18, 41-902 Bytom, Poland

**Keywords:** nutrition knowledge, physical differences, football, athletes, body composition, anthropometry

## Abstract

The optimal body composition for health is an individual trait and is determined by genetic factors, sex, age, somatotype, physical activity, and individual variability. The present study aims to assess how professional football players’ body composition has changed over the training macrocycle in various age groups and to determine the correlation between nutritional awareness and body composition maintenance. Thirty-eight football players participated in the study, with 16 players classified in the younger age group (19–25) and 22 in the older age group (26–31). Using the direct segmented multi-frequency electrical impedance analysis technique, the athletes’ body composition was assessed six times across a training macrocycle made up of preparatory, competitive, and transitional periods. The Sports Nutrition Knowledge Questionnaire was used to evaluate nutrition knowledge. The above correlations show that both younger and older athletes with higher awareness are better able to adjust their nutrition to meet the goals of the preparation period and can achieve greater gains in muscle mass and greater reductions in body fat. According to the study’s results, athletes who are better conscious of their nutritional needs during competition experience less muscle loss and exhibit more consistent body weight and BMI levels. Football players’ body composition suffers detrimental alterations throughout the transition period. Higher body mass, lean body mass content, and skeletal muscle mass are traits of older players. Higher nutritional knowledge reduces the negative modifications of body composition consisting of muscle mass reduction and fat gain. Nutritional knowledge influences the stability of body composition in both age groups during all the analyzed periods: Preparation, competition, and transition.

## 1. Introduction

One of the most popular sports in the world is football, gathering millions of fans in front of TV sets and in stadiums [1,2]. The popularity of the sport raises expectations for players and pushes the bar for sportsmanship constantly [3]. Nutrition for athletes should include the application of nutritional knowledge to a practical nutrition plan that will ensure an adequate supply of energy tailored to physical activity, the occurrence of recovery processes in the body, the improvement of sports performance and proper health and well-being. Proper nutrition as a component of athletes’ training is important because of the strong association of nutritional deficiencies with the risk of injury and trauma and sports performance [3,4]. It is important to measure the body composition of players in order to determine the effectiveness of training and nutrition [4]. Athletes must be able to manage the physical demands, perform at their absolute best, and do so over the full season. As a result, the evaluation of body composition, which establishes an athlete’s level of physical fitness, is crucial to any team’s success [1].

In football, the preparation of players is usually done in a two-cycle model [5]. The spring and fall round systems of competition are a part of this type of training regimen. There are three periods—preparatory, competitive, and transitional—in each macrocycle training round, which represents a single stage in the development of a sport’s form [5,6].

The preparatory period in the sport of football is aimed at developing fitness in preparation for the competitive period [6]. After the transition time, players’ fitness is rebuilt throughout this time period of training. Compared to the starting period, training loads are generally increased [7,8,9]. During this time, technical and tactical drills are organized along with heavy physical training and a string of friendly games [10]. Over the course of the preparation period, under the influence of training stimuli and an appropriate diet, dynamic changes should occur in body mass and body composition primarily with respect to fat mass and lean mass. Athletes’ body mass should increase, mostly owing to an increase in their fat-free mass, and they should simultaneously reduce their fat mass and raise their fat-free mass to maintain their body mass. Athletes’ strength and power increase as a result of gains in fat-free mass [9,10].

The extended competitive period that follows the preparatory stage is marked by the appearance of many match games that require a lot of energy. More match play results in a very high workload for elite players during the competitive period [6]. In order to maintain an optimal body composition during the competitive period, a professional athlete needs to adapt his nutrition to the exercise loads happening throughout this time. Energy intake in the excess of the body’s requirements can result in an increase in body fat, while an inadequate energy supply and an incorrect ratio of macronutrients in the diet can contribute to increased muscle catabolism and a decrease in the athlete’s muscle mass. The risk of injury can be increased in elite football players, which can have a negative effect on performance and health [10,11].

The training is completely stopped or significantly scaled back during the transition period. The kinetics of changes in body composition will be modulated by the length of the transition period, the reduction in training units, and the athletes’ level of fitness, which could result in a partial or total loss of some training-induced adaptations [12]. Reduced training units and club match play, inactive leisure time, and the inability of professional soccer players to adapt their nutrition to reduced physical activity during the transition period may lead to undesirable changes in body mass composition [11,12,13]. The performance and fitness of the athletes are impacted by the increase in body mass brought on by a poor diet. When many and intense training units are reintroduced during the preparatory period, decreasing muscle mass brought on by a lack of training stimulus can lead to decreased strength and endurance, which increases the risk of injury [14].

Assessing body composition throughout the round with periods (preparation, competitive, and transition periods) is extremely important due to the different training intensities, coaching staff care, the number of games played, and time off during each period [11,15].

For football players, optimal body composition is a prerequisite for health and influences athletic performance by affecting movement quality and performance levels [16]. The optimal body composition for health is an individual trait and is determined by genetic factors, sex, age, somatotype, physical activity, and individual variability [17].

Professional athletes need to eat well and get enough nutrients to support their training and increase their physical performance [11]. Professional football players who lack adequate nutritional knowledge may make poor dietary decisions [9]. As a result, there is an imbalanced use of energy, which can lead to weight gain or loss, impaired capacity for exercise, or an elevated risk of trauma and injury [9,11]. Therefore, in order for professional football players to perform well on the field, adequate nutritional knowledge and practices regarding the kind and quantity of food consumed, hydration intake, and timing of consumption are crucial [11].

Adequate nutrition in sports is extremely important, but many athletes make nutritional mistakes [18]. Some authors suggest that the nutritional mistakes found in the athlete population are due to low nutritional awareness [19,20]. Improving dietary habits can positively affect athletic performance [21]. Therefore, increasing nutritional awareness is a key strategy for optimizing sports performance and ensuring proper recovery and long-term health [22,23].

Today’s professional football players are athletes who achieve performance at an extremely high physical level. They manage to do this even though the competition’s schedule is increasingly demanding and tight. However, even with the most modern methods of training, wellness, nutrition, supplementation, and recovery, physiology has its share of limiting players’ performance. A study conducted by Dendir analyzed one aspect of this phenomenon and that is the optimal age for achieving the best athletic form. The study’s results show that the average football player peaks between the ages of 25 and 27. The market value of athletes ranks similarly. Statistically, the highest market values are achieved by athletes at the age of 25 [24]. This age also corresponds to the full ontogenetic development and achievement of maximum skeletal muscle strength that determines body composition [25,26].

According to the research, a player’s ideal body composition is maintained by factors like sex, age, dietary practices, and training loads [27,28,29,30,31]. However, there are not many publications in the available bibliography reporting changes in body composition during the macrocycle, and reports on whether the age of athletes is associated with better body composition management and higher levels of nutritional knowledge. The goal of this study was to ascertain the body composition stability of professional football players while accounting for age and nutritional expertise during the preparation, starting, and transition periods of the spring round of the 2020/2021 football season.

The hypothesis of the study was that older athletes have higher muscle mass content and have higher nutritional awareness and better management of body composition throughout the training macrocycle.

## 2. Materials and Methods

### 2.1. Research Project

The investigation was carried out between 7 January 2021, and 23 July 2021, during the PKO BP Ekstraklasa spring round of the 2020–2021 football season (the highest football competition league in Poland). Male football players from two Silesian clubs competing in the PKO BP Ekstraklasa 2020/2021 made up the study group.

Being a professional football player in one of the clubs involved in the study (the player must sign a professional contract), consenting to participate in the study, and not having any long-term injuries that would prevent them from participating in training and games for the duration of the study, or seven months were the requirements for inclusion in the study.

Football transfers during the study period (7 January 2021 to 23 July 2021), failure to speak Polish or English at a communicative level, absence from practise and game play for at least 14 consecutive days due to illness, injury, isolation, or quarantine due to the COVID-19 pandemic, and missing at least 1 of the 6 measurements for a reason other than those listed above were all exclusion criteria.

The Declaration of Helsinki of the World Medical Association guided the conduct of this study. The Silesian Medical University in Katowice’s Bioethics Committee evaluated and approved the study protocol (PCN/0022/KB/68/I/20). All subjects gave their written, informed consent to take part in the study.

Body composition analyses were carried out six times during preparatory, competitive, and transition periods of the spring round of the 2020/2021 PKO BP Ekstraklasa football competition. Measuring days are scheduled taking into account league and cup matches and providing appropriate time for post-training recovery (according to the test procedure recommended by the InBody producer, the recovery time between body composition analysis and physical activity contributed a minimum of 24 h). Figure 1 shows a diagram of body composition measurements during a training macrocycle.

The study assessed the sports nutrition awareness of all athletes. Due to the presence of non-Polish speaking players in the study group, consultations were conducted in Polish or English language.

### 2.2. Study Group

The 58 football players in the study group ranged in age from 19 to 31. The research included athletes of various nationalities. Figure 2 shows the details.

The classification was made in terms of age: The younger age group consisted of athletes aged 19–25, while the older age group was 26–31. Such a division was made taking into account the optimal age for the development of the best sports performance and the achievement of full ontogenetic development [24,25,26]. The inclusion and exclusion criteria were taken into consideration, and 228 measurements of body composition from 38 athletes were acquired.

### 2.3. BMI Measurement Procedure

Body mass was measured during each body composition analysis, and height was measured before the first body composition analysis.

With a 0.1 cm and 0.1 kg accuracy (SECA 756, Seca gmbh & Co. KG., Hamburg, Germany), height (cm) and body mass (kg) were measured (InBody 770, InBody USA, Cerritos, CA, USA). The contestant was unshod and was only wearing underwear. Body mass (kg) divided by height (m) squared was the method used to compute BMI (Body Mass Index). The findings served as the foundation for evaluating height-to-weight ratios in relation to WHO (World Health Organization) recommendations and guidelines for the European population [32].

### 2.4. Body Composition Analysis Procedure

The Direct Segmental Multi-Frequency Bio-Electrical Impedance Analysis (DSM-BIA) tool was used to calculate body composition (InBody 770, InBody USA, Cerritos, CA, USA). DSM-BIA measures impedance directly from specific bodily compartments and is based on the notion that the human body is made up of five interconnected cylinders. The torso, arms, and legs of the test participant are individually measured for impedance at six different frequencies (1 kHz, 5 kHz, 50 kHz, 250 kHz, 500 kHz, and 1000 kHz) using a tetrapolar eight-point touch electrode system. The tool allows for the measurement of the entire body’s composition in the mid-sixties. An 80 uA current is used by the analyser to operate [33,34].

The parameters of body composition were obtained using Lookin’Body Software version 120.3.0.6. Measurements were carried out in accordance with a set technique and all manufacturer’s instructions. The analyzer was tested with a calibration circuit of known impedance (resistance = 500.0; reactance = 0.1; error = 0.9%) before to each measurement session. The investigation followed accepted practises and supplied relevant literature [33,34,35,36].

For measuring body composition, dual energy absorptiometry (DEXA) is presently the standard technique. However, in order to increase the stability and reproducibility of body composition assessment, BIA technology has substantially advanced to use several frequencies and impedance measurements. Studies among clinical populations, healthy athletes, and physically inactive people have shown that the DSM-BIA method employed in the InBody 770 is valid and reliable when compared to the reference DEXA method [35,36].

### 2.5. Nutrition Awareness Level

The nutritional knowledge of the study group of football players was evaluated using a survey method and the Nutrition Knowledge in Sports Questionnaire (NSKQ) [37,38]. Utilizing both the questionnaire’s original English version and a Polish version created for this study, nutrition knowledge was evaluated. Using an eight-item technique, the NSKQ questionnaire created by Trakman et al. was verified. PerSepIndex, a summary statistic developed by RUMM 2030 and equivalent to Cronbach’s alpha, was used to assess reliability. Multiple split-half reliability assessments serve as the foundation for both PerSepIndex and Cronbach’s alpha; the standard internal reliability threshold is 0.7 [38]. The questionnaire’s original creators granted the author’s study their written consent to use it. For the purposes of this investigation, a pilot study was carried out on a sample of 30 persons using the Polish version of the questionnaire. The purpose of the pilot study was to validate the Polish-language version of the questionnaire and evaluate its content’s applicability and acceptability. Two different English translators converted the questionnaire’s original Polish version from English. Both translators were native speakers of Polish. These translations were used to produce a single Polish-language NSKQ questionnaire, after which all of the questions were discussed until consensus was reached. In the end, a version that satisfied the requirement of semantic consistency for each response was obtained. Other independent translators then produced two back-translations. An English-speaking individual who examined them for consistency with the original questionnaire did so. The survey’s Polish translation was then modified to maintain its graphic consistency with the English original (font, text size, order of the questions and answers, and page count).

The Cronbach’s alpha coefficient for the sample after normalisation was 0.83, demonstrating the high reliability of the questions. According to estimates, the PerSepIndex coefficient for the survey portion of the survey proper was 0.8, which is in line with the Cronbach’s coefficient discovered in the pilot study [37,38]. The survey consisted of 87 questions divided into six subsections: Weight control (n = 12), macronutrients (n = 30), micronutrients (n = 13), sports nutrition (n = 12), supplementation (n = 12), and alcohol (n = 8). Because each section is one-dimensional, each part can be utilised separately to gauge each field’s level of nutritional awareness [39]. There was a 25-min test period. A correct response to each of the 87 questions received one point, and a correct answer to all 87 questions resulted in a score of 100%. Nutrition knowledge was assessed quantitatively using the scoring established by Trakman et al. [37].

To avoid giving inaccurate information, the dietary knowledge survey was carried out between the fifth and sixth body composition assessments under individual consultations.

### 2.6. Statistical Procedures

Under the terms of the GNU GPL licence, data were processed using Statistica 13.3 (StatSoft Polska Sp. z o.o., Cracow, Poland) and the R 4.0.0 package (2020) (The R Foundation for Statistical Computing, Vienna, Austria).

Quantitative data were represented by mean values and standard deviations (XS), whereas qualitative data were examined using percentage notation. The study used measurements of total body composition components. The results were the comparative value used in the statistical tests described.

A normal distribution was tested for conformance using the Shapiro-Wilk test. The Student’s *t*-test was used to determine the significance of the differences between the means in the age groups (19–25 years, 24–31 years).

The Mann-Whitney U test was used to assess the concordance of distributions that deviated from the normal distribution. Tukey’s HSD test for parametric analysis and Dunn’s test for non-parametric analysis were employed as post-hoc testing for group comparisons. Calculations were made for both the average measurement (reported as the average of six measures) for the whole study period as well as each individual measurement.

Depending on how closely the distributions adhered to the normal distribution, an ANOVA analysis for repeated measurements or a non-parametric Friedman test was undertaken for the comparative study of anthropometric measures collected at various dates. Tukey’s HSD test or a post-hoc test for Friedman’s test was used for the suitable post-hoc testing for intergroup comparisons.

An examination of anthropometric measurements (taking into consideration adult measuring norms) and nutrition knowledge (poor, average, good, excellent) in groups by age (19–25 years, 26–31 years) was also done. The association was assessed using Fischer’s test for nxm tables or the 2nd test with changes based on sample size.

The discrepancies between maximum and minimum body composition measures throughout practice, competitive, and transition phases were then analysed for correlation with nutritional awareness. It was assessed how dietary awareness affected the stability of body composition (as determined by the difference between the maximum and minimum measures) over each of the three distinct time periods.

The averaged anthropometric measurements taken during the study periods were employed in the analysis of Spearman’s R’ correlation coefficient with its test of significance regarding the outcomes of the nutrition awareness questionnaire.

Corrections were applied to statistical tests for small subgroups and multiple comparisons.

The criterion for statistical significance was *p* < 0.05.

## 3. Results

During the spring round of the 2020–2021 PKO BP Ekstraklasa season, 6 measurements of 228 body composition were investigated in the current study.

Participants in the study did not disclose any chronic illnesses or medications taken (except one athlete taking Fostex, a drug used to treat asthma, active ingredients beclometasone dipropionate, and formoterol fumarate dihydrate).

### 3.1. Physical Characteristics and Body Composition

Table 1 displays information on age, height, body mass, BMI, and body composition factors such as fat-free mass, skeletal muscle mass, and fat mass. Statistically, significantly higher height, body mass, BMI, fat-free mass content, skeletal muscle mass, percentage of body fat, and fat mass were associated with athletes in the 26–31 age group.

The following phase of the research examined changes in body mass and BMI during the spring round of the 2020–2021 football season’s preparatory, competitive, and transition phases. A separation of the participants into age groups was used to study changes in these factors.

There was statistically significant variance in body mass and BMI across all parameters, including after accounting for age groups. Figure 3 presents the specifics.

Statistically significant variation in the body composition of athletes regardless of age was found in almost all measurements. Larger values of body composition parameters were shown in the group of older athletes. Younger age groups showed statistically significant increases in FFM content between the fifth and sixth assessments (*p* < 0.05), whereas older age groups showed statistically significant increases between the second, third, fourth, and sixth measurements (*p* < 0.01). In measures 1–6, there were differences in the SMM content of the study group that were statistically significant. Between 1 and 2 and 1 and 6 measurements, the SMM of the study group increased (*p* < 0.0001). In the younger age group, SMM was found to rise between measures 1, 4, and 6 (*p* < 0.0001). Increased SMM was found between measurements 1 and 2, 3, 4, and 6 in the older age group (*p* < 0.0001). Analyzing the data on %BF showed statistically significant differences in all measurements. Adipose tissue content [%] increased between 2 and 5 measurements and between 2, 3, 4 and 6 in the older age group (*p* < 0.01). The younger age group showed a statistically significant increase in %BF between 2, 3 and 6 measurements (*p* < 0.05). In measurements 1–6, statistically significant differences were found in the FM of the study group. Between measurements 2, 3, 4, and 6, FM increased (*p* < 0.01). Figure 4 displays the specifics.

### 3.2. Nutrition Knowledge

The majority of dietary knowledge came from expert sources. The main source of information in this area for 60.5% (N = 23) was dietary recommendations. Only a minor number of respondents (N = 12) learned about nutrition via a trainer, compared to as many as 71.1% (N = 27) who learned about it online and 34.2% (N = 13) who learned it from other people.

In the assessment of nutritional knowledge, there were six thematic groups, which included: Weight control, macronutrient and micronutrient content of the diet, sports nutrition, use of supplementation, and alcohol consumption.

Respondents received an average of 51.49% correct answers, and their nutritional awareness was assessed as the average. Athletes scored an average of 48.6% in the section on nutritional knowledge of micronutrients in the diet, compared to 34.6% in the section on supplements. The level of nutritional knowledge in these categories was subpar. Football players scored an average of 52.6% correctly on questions on nutrition and weight control, 60.3% on questions about macronutrients in the diet, 50% on questions about sports nutrition, and 49.9% on questions about alcohol intake. Nutritional awareness in these categories was at an average level.

Nutrition awareness was also verified with the age of the subjects. Detailed information is shown in Figure 5. Age groups’ knowledge did not differ statistically significantly (*p* > 0.05).

There was no variance that was statistically significant in overall nutritional knowledge and nutritional knowledge from each subcategory analyzed among football players by age category (*p* > 0.05).

### 3.3. Modifications in Body Composition Taking into Account the Level of Nutritional Knowledge

The sub-section of nutritional knowledge and body composition throughout the preparatory phase showed statistically significant correlations in the younger age group. According to statistics, a higher SMM dispersion was connected to a considerably higher weight management (r’ = 0.51). Body mass dispersion (r’ = −0.56) and BMI values both had a highly significant negative link with awareness status in the area of sports nutrition (r’ = −0.51). Higher alcohol consumption knowledge had a negative (r’ = −0.56) correlation with LBM content dispersion. There was a statistically significant negative association between awareness of the category of sports nutrition and dispersion of LBM content during the competitive period in the younger age group (r’ = −0.56). Higher knowledge from the category of alcohol consumption was associated with lower dispersion of body mass (r’ = −0.7), BMI (r’ = −0.67), LBM (r’ = −0.58), and SMM (r’ = −0.55). There was a statistically significant inverse relationship (r’ = −0.52) between knowledge status from the category of weight management and the dispersion of BMI during the transition period. Lower PBF dispersion was related to more knowledge in the supplementation category (r’ = −0.53). The results of the analysis are shown in Table 2.

There were statistically significant connections between aspects of dietary knowledge and body composition in the older age group during the preparation period. A statistically significant negative connection between knowledge status and the category of dietary micronutrient content and the dispersion of %BF (r’ = −0.5) and FM (r’ = −0.46) was seen in the older age group during the preparation period. Higher alcohol consumption knowledge was linked to reduced body mass dispersion and BMI during the competitive phase (r’ = −0.42 and r’ = −0.42, respectively). A statistically significant negative association between knowledge in the categories of weight control and body mass dispersion (r’ = −0.51) and BMI values (r’ = −0.49) was seen in the older age group during the transition period. Lower body mass dispersion (r’ = −0.43), BMI values (r’ = −0.45), and SMM (r’ = −0.47) were all linked to higher knowledge of the category of macronutrient content in the diet. Lower PBF dispersion was associated with more awareness of alcohol use (r’ = −0.59). Table 3 displays the analysis’ findings.

## 4. Discussion

Proper body mass is an essential element in maintaining health, and its composition in athletes is influenced by the physical stress caused by professional football [40]. Scientific research emphasizes the role of elements like age, sex, dietary practices, and training loads in preserving athletes’ ideal body mass composition [27,28].

Age is an important determinant of body mass composition; it is related both to individual development and maturity attainment and to the seniority and experience of the player. In elite teams, the age of football players varies. Players after the age of 15 can begin a professional career with professional clubs, while there is no age that determines the end of the career. The oldest players participating in elite competitions are over 40 years old. The seniority of an athlete, and thus the number of training units and matches played by the player, determines the composition of body mass and changes within it [41,42].

The current study showed that the older age group was characterized by higher height (*p* < 0.01), higher body mass (*p* < 0.001), and BMI value (*p* < 0.05). Studies by other authors show that older athletes are characterized by higher body weight and BMI value on average. A study by Nikolaidis showed that body weight and BMI increase with the age of athletes [43].

Analyzing body mass composition, it was found that older athletes were characterized by higher lean body mass (*p* < 0.05), skeletal muscle mass (*p* < 0.05), body fat [%] (*p* < 0.001), and adipose tissue mass (*p* < 0.0001) on average compared to the younger age group. Milsom et al., found that higher lean body mass content was characteristic of older athletes, while differences in body fat content were not significant [44]. In the authors’ study, the differences in body fat content among age groups are statistically significant.

Athletes’ body composition is significantly influenced by the period of the season and their level of nutritional knowledge [45].

In the author’s study, athletes received an average of 51.49% correctly answered questions, and their level of nutritional knowledge was considered average. Age-related differences in nutritional knowledge were not statistically significant. This finding raises concerns because it suggests that the club’s sports nutritionist, the training staff, and the athlete’s personal education regarding nutrition are ineffective. Studies conducted by other authors using the same questionnaire revealed that both male and female athletes competing in different sports lacked adequate dietary understanding.

In a study conducted by Danh et al., with 14 young female volleyball players, athletes received an average of 45.4% correct answers, and their awareness was rated as poor. Female athletes obtained the most correct answers in the category of macronutrient content in the diet, while the least in the subsection of supplementation, the authors’ study showed similar data [46].

Jenner et al., evaluated the nutritional knowledge of 46 professional Australian football players. The players’ nutritional understanding was rated as inadequate and they received an average of 46% of the questions right [47].

In their study, Jagim et al., examined 67 university athletes’ body composition, nutritional awareness, and capacity to predict their needs for macronutrients and energy. The Abridged Sport Nutrition Knowledge Questionaire (ASNKQ) was the instrument used to gauge nutrition awareness. Athletes properly answered 47.9% of the questions on average, whereas the research group’s expertise was judged to be below average. The athletes’ estimates of their demands for energy and carbohydrates were not accurate. According to the study, people who were more knowledgeable about sports nutrition had lower body fat and fat mass percentages. As a result, athletes with a more in-depth understanding of sports nutrition can more effectively alter their macronutrient and energy needs, which affects the maintenance of the optimal body composition [48].

A study by Condto et al., analyzed food intake, nutritional knowledge, and energy availability in 30 elite female Australian football players. On average, 54.4% of correct answers were obtained in the study, which, according to the classification of nutritional awareness assessed by the NSKQ questionnaire, corresponds to average nutritional knowledge. The surveyed population of women was characterized by inadequate intake of carbohydrates, iron, and calcium in relation to current recommendations, in addition, 30% of those surveyed might be at risk for limited energy availability. Increasing the nutritional knowledge of female Australian football players may positively influence adequate nutrient and energy intake [49].

Ample diet and well targeted exercise stimulus both have an impact on how the body changes in composition. The club’s training staff develops the training programme, which is tailored to the abilities and objectives of the squad as well as the needs of the individual players. The coach, assistants, and physical preparation coaches oversee the proper execution of each exercise during training sessions. The training staff and spectators assess the players’ activity and commitment during match play. An essential component of controlling body composition is proper eating. Given the extensive duties of a club’s dietician and the fact that the majority of a player’s meals are taken away from the club, it is critical to evaluate the level of nutritional knowledge among football players, identify areas that require increased nutritional awareness, and determine the effect of knowledge on changes in body composition throughout the season [50,51]. The nutritionist employed by the clubs included in the study did not provide nutritional education to the players; he was a member of the training staff, participated in body composition assessments, and set menus in the hotel kitchens.

The management and maintenance of body mass composition over certain time periods can be positively impacted by raising the dietary awareness of elite football players of all ages. The research issue under consideration had no data in the literature that was available, thus it would be useful as a benchmark.

The research conducted allowed us to partially disprove the hypothesis. The athletes from the older age group were characterized by higher body composition parameters, including muscle mass content, while there were no differences between the groups in the level of nutritional knowledge and body composition management.

### Strengths and Limitations

Despite the popularity of football, there are few academic studies examining changes in football players’ body mass throughout the course of the training macro-cycle. This is one of the work’s advantages. According to an analysis of current scientific publications, no study has addressed the age of players. What is usually analyzed is the composition of body mass in individual single periods of the macro-cycle. Also of importance is the fact that the study group is mostly made up of amateur or semi-professional soccer players, whose training intensity, number of training units, match requirements, and daily physical activity differ from those received by professional soccer players. The present study analyzed the results obtained from professional soccer players in Poland’s top soccer league. The current study used a standardized questionnaire, which is used in studies involving different groups of athletes, this provides additional comparative value to the work. Nowadays, age is an important determinant of the market value of players, the above study can be practically used by the training staffs of sports clubs to select and adjust appropriate strategies to support players at different points of the season in order to maintain their optimal body composition throughout the round which will allow the player to increase his athletic capabilities. According to GoogleScholar’s bibliometric analysis, this is the first study to examine the relationship between nutritional awareness and changes in body composition in football players by age group (as of 9 December 2022).

It is crucial to point out the study’s shortcomings as well. The study group’s size is one of its flaws, however it is important to note that numerous exclusion criteria were established, and one of them was a 14-day ban from training owing to quarantine or isolation because of COVID-19, which was widespread throughout the football season of 2020–2021. The study only included players from two soccer clubs, therefore that is its main drawback—the absence of players from other clubs. However, the individual training micro-cycle developed by the training staff for each club may be an obstacle to a fair comparison of players. In addition, a control group was not included in the study. The study takes into account only male athletes since the sexual maturation of men and women differs and it would be impossible to compare the selected age groups due to the differences involved. The study also used BIA body composition assessment, the accuracy of which depends on a number of factors. It should be highlighted, nevertheless, that the authors did all possible to reduce bias in the research. The study’s assessment of nutrition knowledge only once during the final exam is another drawback. However, it should be mentioned that the athletes were not subjected to nutrition education at the sports club during the study.

## 5. Conclusions

According to the results, players’ body compositions changed independently of age group during the PKO BP Ekstraklasa’s 2020–2021 spring round’s preparation, competitive, and transitional periods, and both groups’ players had average nutritional awareness, according to the classification used. The characteristics of older athletes included higher height, body mass, and BMI values, as well as higher levels of muscle mass and body fat. Taking into account the times of the spring season, the body composition of athletes, regardless of age group, was considerably altered. In all of the examined phases—preparatory, competitive, and transitional—correlations between aspects of athletes’ dietary knowledge and the consistency of their body composition were discovered. Weight management, diet’s macronutrient content, athletes’ nutrition, and alcohol awareness all had an impact on how stable body composition measurements were in both age groups.

## Figures and Tables

**Figure 1 nutrients-15-00705-f001:**
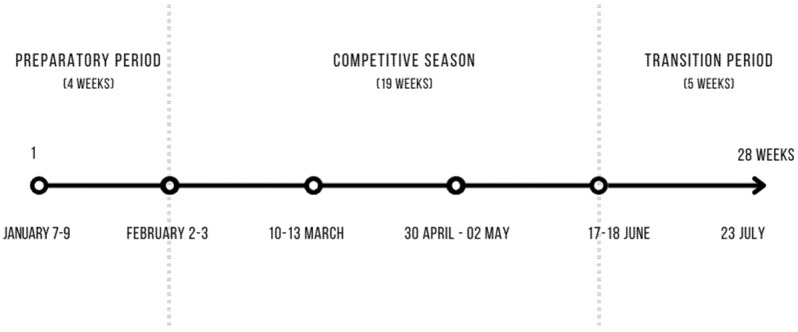
Timing of individual body composition measurements.

**Figure 2 nutrients-15-00705-f002:**
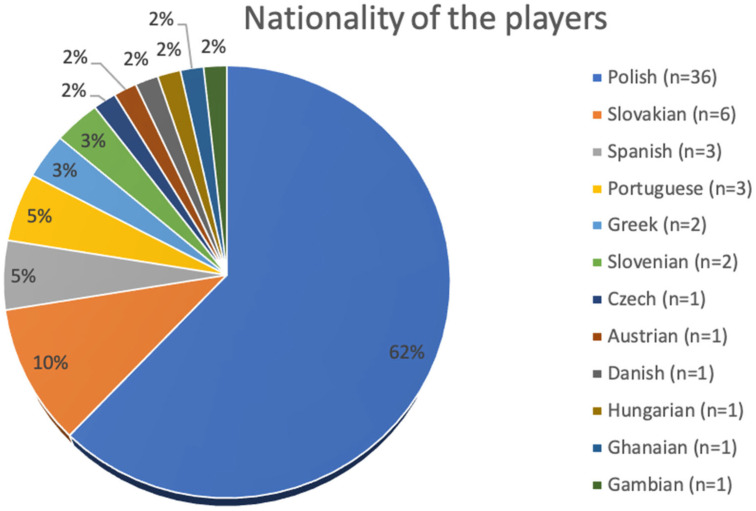
Nationality of the players included in the study.

**Figure 3 nutrients-15-00705-f003:**
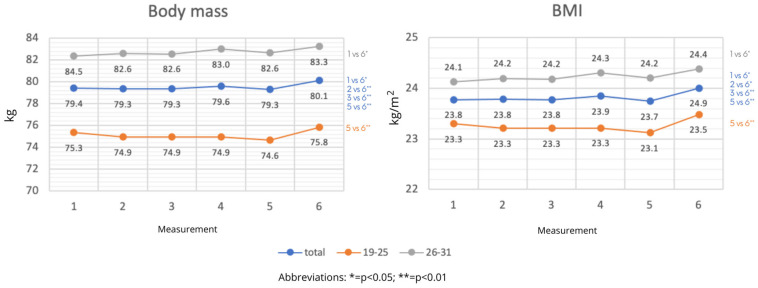
Changes in body weight, and BMI of participants during the training macrocycle by age category (average values).

**Figure 4 nutrients-15-00705-f004:**
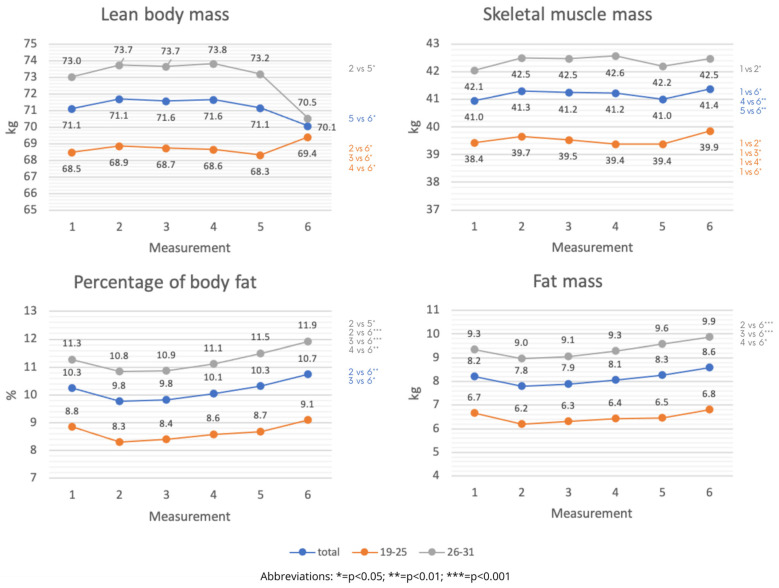
Changes in body composition of participants during the training macrocycle by age category (average values).

**Figure 5 nutrients-15-00705-f005:**
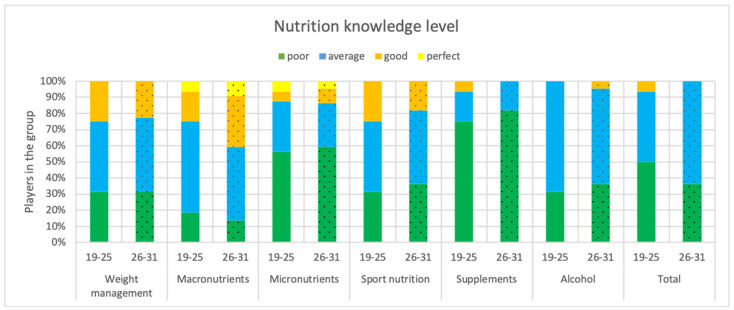
Nutritional awareness of participating players by age category (n = 38).

**Table 1 nutrients-15-00705-t001:** Physical characteristics and variables of body composition.

Variable		Age (Year)	Height (cm)	Body Mass (kg)	BMI (kg/m^2^)	FFM (kg)	SMM (kg)	%BF	FM (kg)
Total (n = 38)		25.9 ± 5.2	182.6 ± 5.5	79.5 ± 7.2	23.8 ± 1.3	71.2 ± 5.8	41.2 ± 3.5	10.2 ± 2.4	8.1 ± 2.4
Age group	19–25 (n = 16)	20.9 ± 2.9	179.8 ± 4.1 **	75.1 ± 3.8 ***	23.3 ± 1.1 *	68.7 ± 3.9 *	39.5 ± 2.5 *	8.7 ± 1.8 **	6.5 ± 1.4 ***
	26–31 (n = 22)	29.5 ± 3.6	184.6 ± 5.5 **	82.7 ± 7.4 ***	24.2 ± 1.2 *	74.0 ± 6.3 *	42.4 ± 3.8 *	11.3 ± 2.3 **	9.4 ± 2.2 ***

Values are presented as mean and standard deviation (X ± SD). FFM = fat free mass; SMM = skeletal muscle mass; %BF = percentage of body fat; FM = fat mass; * = *p* < 0.05; ** = *p* < 0.01; *** = *p* < 0.001.

**Table 2 nutrients-15-00705-t002:** Heat map of the strength of the correlation of differences in maximum and minimum body composition measurements and level on the nutrition knowledge assessment expressed by Spearman’s R’ correlation coefficient in the 19–25 age group. The results are visualized using a false color scale, where red indicates a negative correlation and green a positive one.

Variable	Nutrition Knowledge Sub-Section	Weight Management	Macronutrients	Micronutrients	Sport nutrition	Supplements	Alcohol	Total
Preparatory period	Body mass (kg)	−0.09	0.11	0.17	−0.56 *	0.43	0.13	−0.01
	BMI (kg/m^2^)	0.02	0.05	0.13	−0.51 *	0.47	0.15	−0.01
	LBM (kg)	0.36	0.11	0.04	−0.06	0.15	−0.56 *	0.02
	SSM (kg)	0.51 *	0.22	−0.11	0.09	0.21	−0.11	0.21
	%BF	0.35	−0.3	−0.19	−0.28	−0.13	−0.23	−0.32
	FM (kg)	0.43	−0.15	0.02	−0.24	0.03	−0.23	−0.13
Competitive season	Body mass (kg)	0.13	−0.31	0.03	−0.21	−0.17	−0.7 **	−0.3
	BMI (kg/m^2^)	0.15	−0.37	0	−0.19	−0.21	−0.67 **	−0.35
	LBM (kg)	−0.03	−0.27	−0.01	−0.56 *	0.05	−0.58 *	−0.37
	SSM (kg)	0.11	−0.32	−0.06	−0.28	−0.07	−0.55 *	−0.29
	%BF	−0.02	−0.01	0.1	−0.24	−0.27	−0.45	−0.19
	FM (kg)	0.02	0.02	0.13	−0.17	−0.31	−0.39	−0.15
Transition period	Body mass (kg)	−0.47	0	−0.12	0.12	0.4	0.16	0.13
	BMI (kg/m^2^)	−0.52 *	−0.01	−0.12	0.08	0.35	0.14	0.11
	LBM (kg)	−0.35	0.31	0.28	0.05	0.05	−0.09	0.35
	SSM (kg)	−0.42	−0.13	−0.19	−0.05	0.24	0.16	−0.05
	%BF	−0.08	0.12	0.12	0.23	−0.53 *	0.35	0.42
	FM (kg)	0.03	0.17	0.27	0.16	0.31	−0.02	0.42
	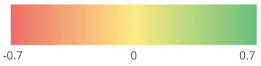							

Abbreviations: LBM = lean body mass; SMM= skeletal muscle mass; %BF = percentage of body fat; FM = fat mass; * = *p* < 0.05; ** = *p* < 0.01.

**Table 3 nutrients-15-00705-t003:** Heat map showing Spearman’s R’ correlation coefficient values for the maximum and minimum body composition measures and level on the nutrition knowledge assessment for population aged 26 to 31. The results are visualized using a false color scale, where red indicates a negative correlation and green a positive one.

Variable	Nutrition Knowledge Sub-Section	Weight Management	Macronutrients	Micronutrients	Sport nutrition	Supplements	Alcohol	Total
Preparatory period	Body mass (kg)	−0.25	0.02	−0.21	−0.03	−0.31	−0.36	−0.27
	BMI (kg/m^2^)	−0.2	0.03	−0.22	−0.01	−0.33	−0.41	−0.26
	LBM (kg)	−0.28	0.2	−0.1	0.31	−0.09	−0.16	0.07
	SSM (kg)	−0.08	0.17	−0.07	0.34	−0.06	−0.23	0.11
	%BF	0.14	0.12	−0.5 *	0.26	0.31	−0.11	0.01
	FM (kg)	0	0.1	−0.46 *	0.14	0.32	−0.12	−0.03
Competitive season	Body mass (kg)	0.02	0.24	0.29	0.08	0.11	−0.42 *	0.31
	BMI (kg/m^2^)	0.04	0.26	0.31	0.11	0.13	−0.42 *	0.35
	LBM (kg)	−0.25	0.06	0.17	−0.37	−0.3	0.01	−0.16
	SSM (kg)	−0.29	−0.08	0.07	−0.4	−0.21	−0.05	−0.29
	%BF	−0.26	−0.26	−0.11	−0.1	−0.18	0.12	−0.21
	FM (kg)	−0.24	−0.28	0.01	−0.09	−0.09	0.29	−0.13
Transition period	Body mass (kg)	−0.51 *	−0.43 *	0.04	−0.31	−0.12	0.16	−0.34
	BMI (kg/m^2^)	−0.49 *	−0.45 *	0.01	−0.31	−0.09	0.12	−0.34
	LBM (kg)	−0.06	−0.09	0.41	−0.02	−0.07	0.26	0.09
	SSM (kg)	−0.04	−0.47 *	−0.06	−0.21	−0.25	−0.06	−0.31
	%BF	0.13	−0.03	0.17	0.16	−0.05	−0.59 ***	0.26
	FM (kg)	−0.17	−0.26	0.09	−0.02	−0.13	0.36	−0.09
	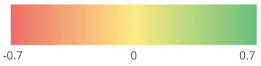							

Abbreviations: LBM = lean body mass; SMM = skeletal muscle mass; %BF = percentage of body fat; FM = fat mass; * = *p* < 0.05; *** = *p* < 0.001.

## Data Availability

The data presented in this study are available on request from the corresponding author. The data are not publicly available due to privacy.

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
