# Peer review of "The Assessment of Body Composition and Nutritional Awareness of Football Players According to Age"

_nutrients, 2023, doi:10.3390/nu15030705_

Round 1

Reviewer 1 Report

Dear Authors,

I am honored to have the opportunity to review the manuscript The Assessment of Body Composition and Nutritional Awareness of Football Players According to Age.

This is a valuable manuscript assessing the body composition of football players in 3 training periods varying in exercise intensity according to age and highlighting the role of nutritional knowledge on the body composition of football players, especially since the literature on this topic is not very extensive. It provides valuable guidance for nutritional strategies to improve body composition and sports performance of football players.

Despite this, the manuscript needs some revisions. Below are some suggestions and questions.

Introduction

It is worthwhile to elaborate on the topic concerning the changes that occur in body composition depending on the training period and how body composition affects sports performance and nutritional knowledge affects body composition. In contrast, line 59 - physical fitness assessment was not analyzed in this study.

Methodology

Research design

In line 127-8 please clarify this, what was the recovery period, Fig 1 shows the timing of the measurements and not the procedure. Was a minimum sample size calculated?

Research group

This would include how many players were in each group and the average age and whether anyone was excluded or dropped out during the study, if so why.

Body composition analysis procedure

Line 178-181 please explain the relationship to the methodology described, line 192-95-just write that the study was conducted according to the current procedure and provide the literature, also simplify the section comparing DEXA and BIA. 

Level of nutrition awareness

How was nutritional knowledge assessed? In the weight and subsequent sections, how did you assess this knowledge? Checking body weight, macronutrients and micronutrients says nothing about the methodology. What were the categories and what was the score in each (poor-perfect)? Who conducted the survey?

Was the survey recorded?

Results

In the description of the results for Fig.4, it would be different to describe the significance of the differences for scores reported greater than 2. Please also check the results for BMI (line 341, 364), FFM (343), PBF (350), BM (364). Negative correlations with a "-" sign? And align the text with the table: FFM, SMM PBF. I have doubts about the color scheme in table 3.

The discussion needs to be organized. Lines 393-394 what research is involved - it is unclear. The discussion avoids repeating the results from the results section in the form of X±SD or p. Isn't line 396-397 a repetition of 393-394. On the other hand, 449-450 and the next sentence are the same? 474-478 please explain it. 

Line 520-523 "disprove the hypothesis"? she also included more muscle mass content in the elderly so maybe modify?

Body mass composition management (line 523)? Please define body mass composition management? Did you mean body mass composition?

Please adapt the references to the requirements of the journal.

Greetings

Author Response

Response to Reviewer 1

Thank you so much for taking the time to evaluate our work. We have tried to incorporate all your valuable suggestions. We have incorporated all the suggestions in the text. Thank you very much. If we could improve our work in any way, please let us know.

Comment 1. It is worthwhile to elaborate on the topic concerning the changes that occur in body composition depending on the training period and how body composition affects sports performance and nutritional knowledge affects body composition. In contrast, line 59 - physical fitness assessment was not analyzed in this study.

Thank you very much for your comment. Porawas as suggested in the introduction of the study.

Comment 2. In line 127-8 please clarify this, what was the recovery period, Fig 1 shows the timing of the measurements and not the procedure. Was a minimum sample size calculated?

According to the test procedure described by the manufacturer of the InBody analyzer and the procedures presented in scientific studies, the time between physical activity and body composition analysis should be a minimum of 24 hours. So we used a minimum 24-hour recovery time before the test. Added this information in the text.

A calculator was used to calculate the ideal sample size based on the following formula.

Nmin=[NP(α2⋅f(1-f))NP⋅e2+α2⋅f(1-f)]/[NP(α2⋅f(1-f))NP⋅e2+α2⋅f(1-f)]

Where,

Nmin - minimum sample size

NP - the size of the population from which the sample is taken - 61 professional football players in the Silesian agglomeration.

α - confidence level for the results, we set at 0.95 (95%)

f - the fraction size - we set at the level of 0,5.

e - assumed maximum error - 0.05

After all calculations, we obtained the result N min = 53 as the required number of subjects in the study.

Fifty-eight players were included in the study, but due to inclusion and exclusion criteria (primarily the timing of the Pandemic COVID-19), the final study material was obtained from 38 players.

Comment 3. This would include how many players were in each group and the average age and whether anyone was excluded or dropped out during the study, if so why.

Information on the number of players in the groups and the average age is described in Table 1. Exclusion criteria describe players who dropped out during the study.

Comment 4. Line 178-181 please explain the relationship to the methodology described, line 192-95-just write that the study was conducted according to the current procedure and provide the literature, also simplify the section comparing DEXA and BIA.

The passage indicated by the reviewer was unnecessary. Thank you very much for your suggestion, it has been removed from the text. The section on the test procedure and the description of the comparison of DEXA and BIA has also been corrected.

Comment 5. How was nutritional knowledge assessed? In the weight and subsequent sections, how did you assess this knowledge? Checking body weight, macronutrients and micronutrients says nothing about the methodology. What were the categories and what was the score in each (poor-perfect)? Who conducted the survey?

The survey was conducted using the NSKQ. The authors of the questionnaire describe in the cited studies the validation and the scoring scale and scores in each section. Section 2.5 cites the studies and indicates that the method of assessment followed the questionnaire authors' guidelines. The survey was conducted in a one-on-one consultation by the authors of the study.

Comment 6. Was the survey recorded?

The NSKQ questionnaire was validated by the questionnaire's authors. In addition, due to the use of the Polish version created for the study, it was also validated. A description is provided in section 3.2.

Comment 7. In the description of the results for Fig.4, it would be different to describe the significance of the differences for scores reported greater than 2. Please also check the results for BMI (line 341, 364), FFM (343), PBF (350), BM (364). Negative correlations with a "-" sign? And align the text with the table: FFM, SMM PBF. I have doubts about the color scheme in table 3.

Thank you very much for all your suggestions. Of course I agree with them, corrected according to the comments.

Comment 8. The discussion needs to be organized. Lines 393-394 what research is involved - it is unclear. The discussion avoids repeating the results from the results section in the form of X±SD or p. Isn't line 396-397 a repetition of 393-394. On the other hand, 449-450 and the next sentence are the same? 474-478 please explain it.

Thank you very much for all your comments. They are due to a clerical error and all have been corrected.

Comment 9. Line 520-523 "disprove the hypothesis"? she also included more muscle mass content in the elderly so maybe modify?

Of course, I agree with the suggestion. Corrected according to the reviewer's comment.

Comment 10. Body mass composition management (line 523)? Please define body mass composition management? Did you mean body mass composition?

Management is a set of activities (including planning and decision-making) for body composition. The athlete should have the ability and knowledge of how to do this.

Comment 10. Please adapt the references to the requirements of the journal.

The references has been corrected as required.

Thank you for your help. Your guidance is invaluable.

Kind regards,

Authors.

Reviewer 2 Report

Line 24-48 – The abstract is very informative, perhaps more than it should be given that is filled with data and stats related to it. There is only one sentence at the end of it that describes what these findings actually mean. My only concern is whether someone from non-scientific community will be able to read the abstract and derive any information from it. I would suggest rephrasing so broader audience can understand the content.

Line 49-51 – Keywords should be different than the title. Please change accordingly.

Line 161 – eligibility criteria should be clearly stated at the beginning of the methods section. Move upwards

Line 165 – BMI is not relevant for athletes… Delete everything related to it.

Line 170 – Full name of the technology and then the acronym in the brackets

Line 173-181 – Not needed. This is already well known.

Line 273-4 – Repetitive. Delete.

Line 304 – Double space. Delete.

Line 351 - Double space. Delete.

Line 352 – Odd positioning of the title of the Table 2. Please adjust so it is on the same page.

Line 382 – 385 – Stated many times through the manuscript. Delete.

Line 422 – Female is repetitive. Delete.

Line 450 – Exclude mass.

The discussion is a bit redundant and I would suggest significant reduction in size. However, paper is overall interesting and unfortunately reaffirms what is already know from other sports. 

Author Response

Response to Reviewer 2

Thank you so much for taking the time to evaluate our work. We have tried to incorporate all your valuable suggestions. We have incorporated all the suggestions in the text. Thank you very much. If we could improve our work in any way, please let us know.

Comment 1. Line 24-48 – The abstract is very informative, perhaps more than it should be given that is filled with data and stats related to it. There is only one sentence at the end of it that describes what these findings actually mean. My only concern is whether someone from non-scientific community will be able to read the abstract and derive any information from it. I would suggest rephrasing so broader audience can understand the content.

Thank you very much for your valuable comments. The summary has been corrected. In its current form, it is certainly very much more readable for the audience. Thank you very much for the valuable guidance.

Comment 2. Line 49-51 – Keywords should be different than the title. Please change accordingly.

Thank you very much for your attention. Changed in the text.

Comment 3. Line 161 – eligibility criteria should be clearly stated at the beginning of the methods section. Move upwards.

Changed according to the comment.

Comment 4. Line 165 – BMI is not relevant for athletes… Delete everything related to it.

Thank you very much for your attention. We decided to leave BMI in the study results because it provides a comparative value to body composition. As the results of our study and the studies of other authors indicate, it is not a reliable indicator for assessing body mass. The BMI index in numerous studies conducted with athletes is described, and of course we agree with the reviewer's opinion, however, we want it to provide informative and comparative value.

Comment 5. Line 170 – Full name of the technology and then the acronym in the brackets

Corrected in the text.

Comment 6. Line 173-181 – Not needed. This is already well known.

Thank you very much for your suggestion. Corrected as per reviewer's comment.

Comment 7. Line 273-4 – Repetitive. Delete.

Line 304 – Double space. Delete.

Line 351 - Double space. Delete.

I am very sorry for the typing errors. Correction.

Comment 8. Line 352 – Odd positioning of the title of the Table 2. Please adjust so it is on the same page.

Thank you very much for all your comments. Corrected in the text.

Comment 9. Line 382 – 385 – Stated many times through the manuscript. Delete.

The indicated passage has been removed.

Comment 10. Line 422 – Female is repetitive. Delete.

Corrected as suggested.

Comment 10. Line 450 – Exclude mass.

Thank you very much for all your comments. Corrected in the text.

Comment 11. The discussion is a bit redundant and I would suggest significant reduction in size. However, paper is overall interesting and unfortunately reaffirms what is already know from other sports.

Thank you very much for your attention. The discussion has been modified.

Thank you for your help. Your guidance is invaluable.

Kind regards,

Authors.

Reviewer 3 Report

The authors provide information on changes in body composition throughout the football season in male athletes and relate that to nutrition knowledge. While the data is of interest, there are too many missing outcomes. Information on dietary intakes, training, access to nutrition support, etc. are all missing. Furthermore, the manuscript is not cohesive and the expected information in the introduction and discussion to support the study is not provided. Finally, the manuscript does in some cases require review for the English language as words are used incorrectly.

Abstract

Dispersion - usually refers to the distribution. I think you are referring to the proportion of skeletal muscle mass or other. As worded it implies there is a correlation between how the muscle is spread throughout the body, not the absolute amount. Correct in results as well. 

Introduction

In general, the connection between the nutrition knowledge and changes in body weight throughout the training cycle requires more information and justification. Presumably the authors believe that nutrition knowledge is stable during the training cycles, as this was only tested once. Therefore the changing variables are body composition and phase. Why should the changes in body composition that occur during the training cycle be related in any way to the athlete’s nutrition knowledge? What is known about changes in body composition throughout the training cycle? What should the changes in body composition be throughout the cycle and how is this related to nutrition knowledge that is being tested here?

Line 55 - It is not clear the relationship between sportsmanship (usually defined as behaviour or treatment of others in sport) and nutrition knowledge? Perhaps consider replacing this content with a more relevant statement regarding the physical/nutrition demands of the sport.

Line 59 - You are discussing the physical fitness, but assessing the nutrition knowledge and body composition. There seems to be a disconnect. Consider information on the importance of body composition assessment to performance or health.

Line 64 - physical skills or fitness or both?

Line 66 - do you mean “competitive period” rather than “starting period”

Line 81 - do you mean “gender” or “sex” - also while you note the importance of this, your sample only includes males - why did you not include females? Also, please discuss the exclusion of the females in the limitations and note that the results cannot be broadly applied. 

As the research hypotheses are stated as you would expect for a single tailed test, however, this is not typically how the data would be analyzed. One should state they expect differences, but not necessarily the direction. Consider revising the hypotheses to align with the methods of analyses or justify the use of a one-tailed test and describe its use in the methods, as it is less common. Furthermore, while it is appropriate to have hypotheses, in the introduction, it is more common to see a study objective or purpose listed. The study justification also needs to be reinforced. 

Methods

How were the athletes approached? Did they have the option not to participate? Was it the whole team or if not what percentage of the team? The information is important, as it is possible that selecting to participate or not resulted in bias.

How were 38 football players determined to be sufficient for the study? 16 and 22 athletes are quite small groups, especially considering the number of comparisons being made. 

The training of the athletes is required. The difference in body composition between the younger and older athletes could be due to differences in training and this is an important factor. Also did all of the athletes in each group do the same training. Finally, how did the training differ in the three phases? Frequency, intensity etc. 

What nutrition training did the athletes receive and what were their sources of information? This should have been included in the data collection.

Figure 1 - It is unclear if the February 2-3 is considered part of the preparatory period (i.e. end) or the competitive period (start) for the analyses. Similar comment regarding June 17-18.

Line 135 - please group all analysis information under a statistics section. This is also confusing: was the body stability the difference between the maximum and minimum during the whole study or the maximum and minimum in each of the study periods? Or within the groups?

Figure 2.2 - Why is this information important; are you conducting analysis by nationality or do you foresee this having an impact on your results? If yes, please conduct some data analysis on this area.

Line 188 - please confirm how long the athletes fasted for (I assume it was not the 24 hours and that only refers to the caffeine or alcohol)?

Body composition - in the results, you discuss skeletal muscle mass; however this is not described in the methods. The methods indicate a measurement of BMC and LBM which includes protein CHO, lipids and soft tissue not isolated skeletal muscle mass, essentially everything except fat mass and bone mass? Clarity is required. 

Good justification of the use of BIA vs DXA.

Be consistent in the DEXA vs DXA abbreviation 

Why was the nutrition knowledge questionnaire not left until after the last test? It is possible that completing the questionnaire caused the athletes to change their behaviours prior to the last body composition test. Please address this in the limitations section. 

Please describe the scoring of the nutrition knowledge test as you use the categories of poor, average etc. in the statistical analyses.

Statistics - provide more detail on the outcomes i.e. did you use total body fat vs lean mass vs muscle mass etc? Did you break it down by body region as the abstract wording implies? Which tests were used for which outcomes?

How did you account for or correct for multiple comparisons?

Results 

Figure 3 and Figure 4 - While one can cross-reference with the methods above to determine which phase the athletes are in it would be good to see training; competition; recovery noted on the figure or in the figure legend as 1-6 does not fully describe the results. Also, typically the descriptor and units go on the Y- axis with a separate title. Usually in English “.” are used to reflect decimals points not “,”. Figure 3 description should also include the statistical tests completed, especially as it is not indicated which data are normally distributed and thus which of the list of statistical tests noted in the methods were applied to which type of data. Also a correction for multiple comparisons should be included. 

Line 308 - What is the 2.5 measurement? Or should it be 2,5, and 6? Same with 3.4? Also is this for the group as a whole or the younger or older group?

Line 310 - Technically a “difference” is always statistically significant or it cannot be called a difference; it is more helpful to know if the difference was an increase or decrease.

Figure 4 - the change in body composition (average) requires explanation. Change from what and how was this calculated? Clarity here and in the methods.

Discussion

Line 382 Author’s should be authors’ unless there is only one author.

While the comparison to other studies regarding the differences in body composition between the older and younger athletes is well done, it is not clear why this is an important factor to study. Furthermore, there is no discussion regarding target body composition for these athletes groups, especially as it relates to health and performance and how the current results align with recommendations. Are these athlete lower than recommended, in range etc. Provide some information on body composition and performance to support your case. Essentially, why does it matter that there are differences in body composition between these two groups? Why study it at all?

Line 409 - please reference the statement that nutrition awareness is an important determinant of an athlete’s body mass composition. The literature is quite varied here, as there are other barriers to putting nutrition knowledge into practice (especially among young male football players who maybe aren’t cooking their own meals).

Why does the fact that there were no differences in nutrition knowledge by age group indicate that the nutrition education provided by the club etc. is ineffective. Perhaps more detail regarding the clubs and the training they provide and the access to nutrition education for the athletes could be described for more context?

Line 422 - Why is the information on the nutrition knowledge of female athletes relevant to your study as you did not include any females?

The whole discussion on nutrition knowledge in athletes does provide a good overview of what others have found, however, it does not connect this to the training phases or changes in body composition, which is the focus of this study. 

Line 449-461 - This information should be supported by a note of whether these specific athletes had access to a dietitian and the level of support provided.

Line 462 - The information regarding body composition change etc. here is relevant, however, please provide more detail on the “athletes” what sport and what age is “younger” etc.

Line 492-496 - Is this data from other studies that should be referenced or from your study? Provide clarity.

While the connection between age and market value is made; the link to age and body composition is not clear. While age can be a determinant in body composition, the effects in this “older” group would likely be minimal. Sarcopenia isn’t typically a significant concern in  26-31 year olds and cardiac capacity usually peaks around mid-thirties. Perhaps this could be justified if there was evidence that body composition at this age is significantly affected by age or muscle mass etc. (again this comes back to the content in the introduction).  

Author Response

Response to Reviewer 3

Thank you so much for taking the time to evaluate our work. We have tried to incorporate all your valuable suggestions. We have incorporated all the suggestions in the text. Thank you very much. If we could improve our work in any way, please let us know.

Comment 1. Dispersion - usually refers to the distribution. I think you are referring to the proportion of skeletal muscle mass or other. As worded it implies there is a correlation between how the muscle is spread throughout the body, not the absolute amount. Correct in results as well.

Dispersion, as defined, is the variation in the observed values of a variable, the basic (next to central tendency) characteristic of a statistical sample. Dispersion is greater the more these values deviate from the central tendency.

In the study presented here, we wanted to address the overall content of specific body mass paramters. The study did not cover the distribution of these parameters across body parts vs. differences in total content, which is why the word dispersion was used.

Comment 2. In general, the connection between the nutrition knowledge and changes in body weight throughout the training cycle requires more information and justification. Presumably the authors believe that nutrition knowledge is stable during the training cycles, as this was only tested once. Therefore the changing variables are body composition and phase. Why should the changes in body composition that occur during the training cycle be related in any way to the athlete’s nutrition knowledge? What is known about changes in body composition throughout the training cycle? What should the changes in body composition be throughout the cycle and how is this related to nutrition knowledge that is being tested here?

Thank you very much for your insightful question. Nutritional knowledge is, of course, one of the components that can determine body composition, according to the authors. According to the current literature review, no study to date has analyzed the effect of nutritional awareness on body composition much less on body composition changes throughout the round. The results obtained in the study show that there is a relationship between these variables, which may be a suggestion to other researchers. I agree with the reviewer's comment that this issue was not sufficiently discussed in the introduction of the study, so the introduction was supplemented with this information. Nutritional knowledge was tested in the final stage of the study, between the 5th-6th measurements. Respondents were not subjected to nutrition education in and out of the club, so knowledge was examined only once.

Comment 3. Line 55 - It is not clear the relationship between sportsmanship (usually defined as behaviour or treatment of others in sport) and nutrition knowledge? Perhaps consider replacing this content with a more relevant statement regarding the physical/nutrition demands of the sport.

Thank you very much for your attention. Added a relevant explanatory passage in the text.

Comment 4. Line 59 - You are discussing the physical fitness, but assessing the nutrition knowledge and body composition. There seems to be a disconnect. Consider information on the importance of body composition assessment to performance or health.

Thank you very much for your suggestion. Corrected as per reviewer's comment.

Comment 5. Line 64 - physical skills or fitness or both?

Corrected in the text. It was about physical fitness.

Comment 6. Line 66 - do you mean “competitive period” rather than “starting period”

I apologize for the typing error. Corrected in the text.

Comment 7. Line 81 - do you mean “gender” or “sex” - also while you note the importance of this, your sample only includes males - why did you not include females? Also, please discuss the exclusion of the females in the limitations and note that the results cannot be broadly applied.

Thank you very much for your valuable comment. Changed "gender" to "sex" in the text. Only male athletes were qualified for the study because the sexual maturation of men and women is different. The age group division that was used in the study would be incorrect. However, we are planning an analogous study on a group of women, which will also address the importance of nutritional awareness in the context of body composition management. An appropriate description has been added in the study's limitations.

Comment 8. As the research hypotheses are stated as you would expect for a single tailed test, however, this is not typically how the data would be analyzed. One should state they expect differences, but not necessarily the direction. Consider revising the hypotheses to align with the methods of analyses or justify the use of a one-tailed test and describe its use in the methods, as it is less common. Furthermore, while it is appropriate to have hypotheses, in the introduction, it is more common to see a study objective or purpose listed. The study justification also needs to be reinforced.

Thank you very much for your attention. Obviously applied as suggested from miany in the description of the hypothesis.

Comment 9. How were the athletes approached? Did they have the option not to participate? Was it the whole team or if not what percentage of the team? The information is important, as it is possible that selecting to participate or not resulted in bias.

The survey was voluntary. The boards of the sports clubs were asked to conduct the study, and after receiving permission, all players signed a statement that they agreed to participate in the study and were informed that their participation was voluntary.

Comment 10. How were 38 football players determined to be sufficient for the study? 16 and 22 athletes are quite small groups, especially considering the number of comparisons being made.

A calculator was used to calculate the ideal sample size based on the following formula.

Nmin=[NP(α2⋅f(1-f))NP⋅e2+α2⋅f(1-f)]/[NP(α2⋅f(1-f))NP⋅e2+α2⋅f(1-f)]

Where,

Nmin - minimum sample size

NP - the size of the population from which the sample is taken - 61 professional football players in the Silesian agglomeration.

α - confidence level for the results, we set at 0.95 (95%)

f - the fraction size - we set at the level of 0,5.

e - assumed maximum error - 0.05

After all calculations, we obtained the result N min = 53 as the required number of subjects in the study.

Fifty-eight players were included in the study, but due to inclusion and exclusion criteria (primarily the timing of the Pandemic COVID-19), the final study material was obtained from 38 players.

Comment 10. The training of the athletes is required. The difference in body composition between the younger and older athletes could be due to differences in training and this is an important factor. Also did all of the athletes in each group do the same training. Finally, how did the training differ in the three phases? Frequency, intensity etc.

Differences in athletes' body composition can of course be attributed to training seniority, but changes over a macrocycle and the ability to manage body composition are no longer so obvious. The athletes in the study belonged to 2 sports clubs and were first-team players. Athletes who did not play in match play were required to participate in compensation matches. The training microcycle of both teams was analagous. These factors made it possible to compare the analyzed parameters.

Comment 11. What nutrition training did the athletes receive and what were their sources of information? This should have been included in the data collection.

Information on knowledge acquisition was included in the text.

Knowledge of nutrition was significantly derived from professional sources.  Dietary advice was the primary place for 60.5% (N=23) to gain knowledge in this area. A small group (N=12) of respondents obtained nutrition knowledge from a trainer, as many as 71.1% (N=27) from the Internet, and 34.2% (N=13) from other people.

Comment 12. Figure 1 - It is unclear if the February 2-3 is considered part of the preparatory period (i.e. end) or the competitive period (start) for the analyses. Similar comment regarding June 17-18.

The survey was designed to measure between the start and end of each period. Therefore, the graph shows the data in this way.

Comment 13. Line 135 - please group all analysis information under a statistics section. This is also confusing: was the body stability the difference between the maximum and minimum during the whole study or the maximum and minimum in each of the study periods? Or within the groups?

The passage describing the measurements was clarified according to the comment.

Comment 14. Figure 2.2 - Why is this information important; are you conducting analysis by nationality or do you foresee this having an impact on your results? If yes, please conduct some data analysis on this area.

The information on nationality was provided to indicate the fact that the survey needs to be conducted in two languages and the possibility of possible exclusion from the survey due to not speaking Polish or English. If you suggest removing this figure please let me know.

Comment 15. Line 188 - please confirm how long the athletes fasted for (I assume it was not the 24 hours and that only refers to the caffeine or alcohol)?.

As suggested by another reviewer, this passage has been removed. The athletes were analyzed in the morning, with the last meal consumed before bedtime with the note that the time between measurement and meal must be at least 8h (according to the test procedure suggested by the InBody manufacturer).

Comment 16. L Body composition - in the results, you discuss skeletal muscle mass; however this is not described in the methods. The methods indicate a measurement of BMC and LBM which includes protein CHO, lipids and soft tissue not isolated skeletal muscle mass, essentially everything except fat mass and bone mass? Clarity is required.

As noted by other reviewers, this section has been removed because the description of the method and method of measurement is described by the manufacturer of the InBody device and is well known.

Comment 17. Good justification of the use of BIA vs DXA.

Thank you very much for your comment..

Comment 18. Be consistent in the DEXA vs DXA abbreviation

Attention was drawn during proofreading and corrected.

Comment 19. Why was the nutrition knowledge questionnaire not left until after the last test? It is possible that completing the questionnaire caused the athletes to change their behaviours prior to the last body composition test. Please address this in the limitations section.

A relevant passage in the study's limitations has been added.

Comment 20. Please describe the scoring of the nutrition knowledge test as you use the categories of poor, average etc. in the statistical analyses.

Section 2.5 describes how nutrition knowledge is assessed. The method used was developed by the authors of the questionnaire. The information is included in the description and the source is cited for readers interested in this issue..

Comment 21. Statistics - provide more detail on the outcomes i.e. did you use total body fat vs lean mass vs muscle mass etc? Did you break it down by body region as the abstract wording implies? Which tests were used for which outcomes?.

The study used measurements of total body composition components. The results were the comparative value used in the statistical tests described. Suggested information was added in the description of the statistical methods used.

Comment 22. How did you account for or correct for multiple comparisons?

For this purpose, a Bonferroni adjustment was applied, that is, dividing the assumed significance level by the number of multiple comparisons. Included in the text.

Comment 23. Figure 3 and Figure 4 - While one can cross-reference with the methods above to determine which phase the athletes are in it would be good to see training; competition; recovery noted on the figure or in the figure legend as 1-6 does not fully describe the results. Also, typically the descriptor and units go on the Y- axis with a separate title. Usually in English “.” are used to reflect decimals points not “,”. Figure 3 description should also include the statistical tests completed, especially as it is not indicated which data are normally distributed and thus which of the list of statistical tests noted in the methods were applied to which type of data. Also a correction for multiple comparisons should be included.

Thank you very much for your suggestion. I have corrected the figures according to the guidelines. Used periods instead of commas and described the "y" axis.

Comment 24. Line 308 - What is the 2.5 measurement? Or should it be 2,5, and 6? Same with 3.4? Also is this for the group as a whole or the younger or older group?

I am very sorry for the typing error. Corrected in the text.

Comment 25. Line 310 - Technically a “difference” is always statistically significant or it cannot be called a difference; it is more helpful to know if the difference was an increase or decrease.

Thank you very much for your comment.

Comment 26. Figure 4 - the change in body composition (average) requires explanation. Change from what and how was this calculated? Clarity here and in the methods.

The methodology of the statistical section describes in detail how the averages used in the results were calculated.

Comment 27. Line 382 Author’s should be authors’ unless there is only one author.

Corrected as suggested.

Comment 28. While the comparison to other studies regarding the differences in body composition between the older and younger athletes is well done, it is not clear why this is an important factor to study. Furthermore, there is no discussion regarding target body composition for these athletes groups, especially as it relates to health and performance and how the current results align with recommendations. Are these athlete lower than recommended, in range etc. Provide some information on body composition and performance to support your case. Essentially, why does it matter that there are differences in body composition between these two groups? Why study it at all.

Thank you very much for the right suggestion. I have corrected the discussion and made the changes.

Comment 29. Line 409 - please reference the statement that nutrition awareness is an important determinant of an athlete’s body mass composition. The literature is quite varied here, as there are other barriers to putting nutrition knowledge into practice (especially among young male football players who maybe aren’t cooking their own meals).

The passage quoted before the reviewer refers to the reviewer's own research. No citation was applied to it, so it is written on the basis of the results obtained in the study.

Comment 30. Why does the fact that there were no differences in nutrition knowledge by age group indicate that the nutrition education provided by the club etc. is ineffective. Perhaps more detail regarding the clubs and the training they provide and the access to nutrition education for the athletes could be described for more context?

The level of nutritional knowledge of the study group did not increase with age. The athlete, along with the training internship, receives more dietary consultations and nutrition education. If the number of these does not affect the level of knowledge then it is probably ineffective

Comment 31. Line 422 - Why is the information on the nutrition knowledge of female athletes relevant to your study as you did not include any females?

In the study mentioned in the discussion, the research tool was the same questionnaire used in your own study. For this reason, the study was cited.

Comment 32. The whole discussion on nutrition knowledge in athletes does provide a good overview of what others have found, however, it does not connect this to the training phases or changes in body composition, which is the focus of this study.

As suggested, the text of the discussion was corrected and the reviewer's comments were taken into account.

Comment 33. Line 449-461 - This information should be supported by a note of whether these specific athletes had access to a dietitian and the level of support provided.

Completed in the text.

Comment 34. Line 462 - The information regarding body composition change etc. here is relevant, however, please provide more detail on the “athletes” what sport and what age is “younger” etc..

Corrected according to the comment.

Comment 35. Line 492-496 - Is this data from other studies that should be referenced or from your study? Provide clarity.

Changed the discussion according to other comments, the passage was previously removed.

Comment 36. While the connection between age and market value is made; the link to age and body composition is not clear. While age can be a determinant in body composition, the effects in this “older” group would likely be minimal. Sarcopenia isn’t typically a significant concern in  26-31 year olds and cardiac capacity usually peaks around mid-thirties. Perhaps this could be justified if there was evidence that body composition at this age is significantly affected by age or muscle mass etc. (again this comes back to the content in the introduction). 

The main objective of the study is to assess nutrition awareness and its impact on body composition management. Thank you very much for your comment.

Thank you for your help. Your guidance is invaluable.

Kind regards,

Authors.

Round 2

Reviewer 3 Report

I thank their authors for their comments and the revised manuscript is improved.